# Learning Optimal Predictive Checklists

**Haoran Zhang**
Massachusetts Institute of Technology
`haoranz@mit.edu`

**Quaid Morris**
Memorial Sloan Kettering Cancer Center
`morrisq@mskcc.org`

**Berk Ustun**[*]
UC San Diego
`berk@ucsd.edu`

**Maryzeh Ghassemi**[*]
Massachusetts Institute of Technology
`mghassem@mit.edu`

## Abstract

Checklists are simple decision aids that are often used to promote safety and reliability in clinical applications. In this paper, we present a method to learn checklists for clinical decision support. We represent predictive checklists as discrete linear classifiers with binary features and unit weights. We then learn globally optimal predictive checklists from data by solving an integer programming problem. Our method allows users to customize checklists to obey complex constraints, including constraints to enforce group fairness and to binarize real-valued features at training time. In addition, it pairs models with an optimality gap that can inform model development and determine the feasibility of learning sufficiently accurate checklists on a given dataset. We pair our method with specialized techniques that speed up its ability to train a predictive checklist that performs well and has a small optimality gap. We benchmark the performance of our method on seven clinical classification problems, and demonstrate its practical benefits by training a short-form checklist for PTSD screening. Our results show that our method can fit simple predictive checklists that perform well and that can easily be customized to obey a rich class of custom constraints.

## 1 Introduction

Checklists are simple tools that are widely used to assist humans when carrying out important tasks or making important decisions [8, 13, 18, 35, 44, 50, 60, 61, 64, 65, 66]. These tools are often used as predictive models in modern healthcare applications. In such settings, a checklist is a set of Boolean conditions that predicts a condition of interest – e.g., a list of symptoms flag a patient for a critical illness when $M$ out $N$ symptoms are checked. These kinds of "predictive checklists" are often used for clinical decision support because they are easy to use and easy to understand [33, 54]. In contrast to other kinds of predictive models, clinicians can easily scrutinize a checklist, and make an informed decision as to whether they will adopt it. Once they have decided to use a checklist, they can integrate the model into their clinical workflow without extensive training or technology (e.g., as a printed sheet [52]).

Considering these benefits, one of the key challenges in using predictive checklists in healthcare applications is finding a reliable way to create them [33]. Most predictive checklists in medicine are either hand-crafted by panels of experts [30, 41], or built by combining statistical techniques and heuristics [e.g., logistic regression, stepwise feature selection, and rounding 39]. These approaches

---

\* Equal supervision.

make it difficult to develop checklists that are sufficiently accurate – as panel or pipeline will effectively need to specify a model that performs well under stringent assumptions on model form. Given the simplicity of the model class, it is entirely possible that some datasets may never admit a checklist that is sufficiently accurate to deploy in a clinical setting – as even checklists that are accurate at a population-level may perform poorly on a minority population [57, 70].

In this paper, we introduce a machine learning method to learn checklists from data. Our method is designed to streamline the creation of predictive checklists in a way that overcomes specific challenges of model development in modern healthcare applications. Our method solves an integer programming problem to return the most accurate checklist that obeys user-specified constraints on model form and/or model performance. This approach is computationally challenging, but provides specific functionality that simplifies and streamlines model development. First, it learns the most accurate checklist by optimizing exact measures of model performance (i.e., accuracy rather than a convex surrogate measure). Second, it seeks to improve the performance of checklists by adaptively binarizing features at training time. Third, it allows practitioners to train checklists that obey custom requirements on model form or on prediction, by allowing them to encode these requirements as constraints in the optimization problem. Finally, it provides practitioners with an optimality gap, which informs them when a sufficiently accurate checklist does not exist.

The main contributions of this paper are:

1. We present a machine learning method to learn checklists from data. Our method allows practitioners to customize models to obey a wide range of real-world constraints. In addition, it pairs checklists with an optimality gap that can inform practitioners in model development.

2. We develop specialized techniques that improve the ability of our approach to train a checklist that performs well, and to pair this model with a small optimally gap. One of these techniques can be used to train checklists heuristically.

3. We conduct a broad empirical study of predictive checklists on clinical classification datasets [24, 32, 37, 51, 58]. Our results show that predictive checklists can perform as well as state-of-the-art classifiers on some datasets, and that our method can provide practitioners with an optimality gap to flag when this is not the case. We highlight the ability of our method to handle real-world requirements through applications where we enforce group fairness constraints on a mortality prediction task, and where we build a short-form checklist to screen for PTSD.

4. We provide a Python package to train and customize predictive checklists with open-source and commercial solvers, including CBC [29] and CPLEX [20] (see https://github.com/MLforHealth/predictive_checklists).

## 2   Related Work

The use of checklist-style classifiers – e.g., classifiers that assign coefficients of $\pm 1$ to Boolean conditions – dates to the work of Burgess [9]. The practice is often referred to as "unit weighting," and is motivated by observations that it often performs surprisingly well [see e.g.,  7, 19, 23, 27].

Our work is strongly related to methods that learn sparse linear classifiers with small integer coefficients [see e.g., 4, 11, 31, 38, 68, 69, 75]. We learn checklists from data by solving an integer program. Our problem can be viewed as a special case of an IP proposed by Ustun and Rudin [68], Zeng et al. [77] to learn sparse linear classifiers with small integer coefficients. The problem that we consider is considerably easier to solve from a computational standpoint because it restricts coefficients to binary values and does not include $\ell_0$-regularization. Here, we make use of this "slack" in computation to add functionality that improves the performance of checklists, namely: (1) constraints to binarize continuous and categorical features into items that can be included in a checklist [see also 10, 12, 71]; and (2) develop specialized techniques to reduce computation during the learning process.

Checklists are $M$-of-$N$ rules — i.e., classifiers that predict $y = +1$ when $M$ out of $N$ conditions are true. Early work in machine learning used $M$-of-$N$ rules in an auxiliary capacity – e.g., to explain the predictions of neural networks [67], or to use as features in decision trees [78]. More recent work has focused on learning $M$-of-$N$ rules for standalone prediction – e.g. Chevaleyre et al. [16] describe a method to learn $M$-of-$N$ rules by rounding the coefficients of linear SVMs to $\{-1, 0, 1\}$ using a randomized rounding procedure [see also 79]. Methods to learn disjunctions and conjunctions are also relevant since they correspond to 1-of-$N$ rules and $N$-of-$N$ rules, respectively. These methods

learn models by solving a special case of the ERM problem in (1) where $M = 1$ or $M = N$. Given that this problem is NP-hard, existing methods often reduce computation by using algorithms that return approximate solutions – e.g., simulated annealing [72], set cover [48, 49], or ERM with a convex surrogate loss [21, 22, 46, 47]. These approaches could be used to learn checklists by solving the ERM problem in (1). However, they would not be able to handle the discrete constraints needed for binarization and customization. More generally, they would not be guaranteed to recover the most accurate checklists for a given dataset, nor pair models with a certificate of optimality that can be used to inform model development.

## 3 Methodology

We start with a dataset of $n$ examples $(\boldsymbol{x}_i, y_i)_{i=1}^n$ where $\boldsymbol{x}_i = [x_{i1}, \ldots, x_{id}] \in \{0, 1\}^d$ is a vector of $d$ binary variables and $y_i \in \{0, 1\}$ is a label. Here, $\boldsymbol{x}_i$ are Boolean variables that could be used as items in a checklist. We denote the indices of positive and negative examples as $I^+ = \{i \mid y_i = +1\}$ and $I^- = \{i \mid y_i = -1\}$ respectively, and let $n^+ = |I^+|$ and $n^- = |I^-|$. We denote the set of positive integers up to $k$ as $[k] = \{1, \ldots, k\}$.

We use the dataset to learn a *predictive checklist* – i.e., a Boolean threshold rule that predicts $\hat{y} = +1$ when at least $M$ of $N$ items are checked. We represent a checklist as a linear classifier with the form:

$$\hat{y}_i = \text{sign}(\boldsymbol{\lambda}^\top \boldsymbol{x}_i \geq M).$$

Here $\boldsymbol{\lambda} = [\lambda_1, \ldots, \lambda_d] \in \{0, 1\}^d$ is a coefficient vector and $\lambda_j = 1$ iff the checklist contains item $j$. We denote the number of items in a checklist with coefficients $\boldsymbol{\lambda}$ as $N = \sum_{j=1}^d \lambda_j$, and denote the threshold number of items that must be checked to assign a positive prediction as $M$. We learn checklists from data by solving an empirical risk minimization problem with the form:

$$
\begin{aligned}
\min_{\boldsymbol{\lambda}, M} \quad & l(\boldsymbol{\lambda}, M) + \epsilon_N N + \epsilon_M M \\
\text{s.t.} \quad & N = \|\boldsymbol{\lambda}\| \\
& M \in [N] \\
& \boldsymbol{\lambda} \in \{0, 1\}^d
\end{aligned}
\tag{1}
$$

Here, $l(\boldsymbol{\lambda}, M) = \sum_{i=1}^n \mathbb{1}[y_i \neq \hat{y}_i]$ counts the number of mistakes of a checklist with parameters $\boldsymbol{\lambda}$ and $M$. The parameters $\epsilon_N > 0$ and $\epsilon_M > 0$ are small penalties used to specify lexicographic preferences. We set $\epsilon_N < \frac{1}{nd}$ and $\epsilon_M < \frac{\epsilon_N}{d}$. These choices ensure that optimization will return the most accurate checklist, breaking ties between checklists that are equally accurate to favor smaller $N$, and breaking ties between checklists that are equally accurate and sparse to favor smaller $M$. Smaller values of $N$ and $M$ are preferable, as checklists with smaller $N$ users check fewer items, and checklists with smaller $M$ let users stop checking as soon as $M$ items are checked (see Figure 3).

We recover a globally optimal solution to (1) by solving the integer program (IP):

$$
\begin{aligned}
\min_{\boldsymbol{\lambda}, z, M} \quad & l^+ + W^- l^- + \epsilon_N N + \epsilon_M M \\
\text{s.t.} \quad & B_i z_i \geq M - \sum_{j=1}^d \lambda_j x_{i,j} && i \in I^+ && \text{(2a)} \\
& B_i z_i \geq \sum_{j=1}^d \lambda_j x_{i,j} - M + 1 && i \in I^- && \text{(2b)} \\
& l^+ = \sum_{i \in I^+} z_i \\
& l^- = \sum_{i \in I^-} z_i \\
& N = \sum_{j=1}^d \lambda_j \\
& M \in [N] \\
& z_i \in \{0, 1\} && i \in [n] \\
& \lambda_j \in \{0, 1\} && j \in [d]
\end{aligned}
$$

Here, $l^+$ and $l^-$ are variables that count the number of mistakes on positive and negative examples. The values of $l^+$ and $l^-$ are computed using the mistake indicators $z_i = \mathbb{1}[\hat{y}_i \neq y_i]$, which are set to

1 through "Big-M" Constraints in (2a) and (2a).These constraints depend on the "Big-M" parameters $B_i$, which can be set to its tightest possible value $\max_\lambda M - \lambda^\top \boldsymbol{x}_i$. $W^-$ is a user-defined parameter that reflects the relative cost of misclassifying a negative example. By default, we set $W^- = 1$, so that the objective minimizes training error.

**Customization**   The IP in (2) can be customized to fit checklists that obey a wide range of real-world requirements on performance and model form. We list examples of constraints in Table 1. Our method provides two ways of handling classification problems with different costs of False Negatives (FNs) and False Positives (FPs). First, the user can specify the relative misclassification costs for FPs and FNs in the objective function by setting $W^-$. Here, optimizing (2) corresponds directly to minimizing the weighted error. Second, the user can instead specify limits on FPR and FNR - e.g. minimize the training FPR (by setting $W^- = n^-$) subject to training FNR $\leq 20\%$.

| Model Requirement | Example | Constraint |
|---|---|---|
| Model Size | Use $\leq N_{max}$ items | $N \leq N_{max}$ |
| Procedural | If checklist includes item about coughing, then include item about fever | $\lambda_{fever} \geq \lambda_{cough}$ |
| Prediction | Predict $\hat{y}_i = +1$ for all patients with both fever and coughing | $\lambda_{fever} x_{i,fever} + \lambda_{cough} x_{i,cough} \geq M \ \forall \, i \in [n]$ |
| Class-Based Accuracy | Max FPR $\leq \gamma$ | $l^- \leq \lceil \gamma \cdot n^- \rceil$ |
| Group Fairness | Max FPR disparity of $\gamma\%$ between males and females | $\left\lvert \frac{l_M^-}{n_M^-} - \frac{l_F^-}{n_F^-} \right\rvert \leq \gamma$ |
| Minimax Fairness | No group with FNR worse than $\delta$ | $l_g^+ \leq \lceil \delta \cdot n_g^+ \rceil \ \ \forall g \in G$ |

**Table 1:** Model requirements that can be addressed by our method. Each requirement can be directly encoded into IP (2). The IP can then be solved using the same MIP solver to obtain the most accurate checklist that obeys these constraints.

**Adaptive Binarization**   Practitioners often work with datasets that contain real-valued and categorical features. In such settings, rule-learning methods often require practitioners to binarize such features before training. Our approach allows practitioners to binarize features into items during training. This allows all items to be binarized in a way that maximizes accuracy. This approach requires practitioners to first define $T_j$ candidate items for each non-binary feature $u_{i,j}$. For example, given a real-valued feature $u_{i,j} \in \mathbb{R}$, the set of candidate items could take the form $\{x_{i,j,1}, x_{i,j,2}, \ldots, x_{i,j,T_j}\}$ where $x_{i,j,t} = \mathbb{1}\left[u_{i,j} \geq v_{j,t}\right]$ and $v_{j,t}$ is a threshold. We would add the following constraint to IP (2) to ensure that the checklist only uses one of the $T_j$ items: $\sum_{t \in [T_j]} \lambda_{t_j} \leq 1$. In this way, the IP would then choose a binary item that is aligned with the goal of maximizing accuracy. In practice, the set of candidate items can be produced automatically [e.g., using binning or information gain 53] or specified on the basis of clinical standards (as real-valued features like blood pressure and BMI have established thresholds). This approach is general enough to allow practitioners to optimize over all possible thresholds, though this may lead to overfitting. Our experiments in Section 5.2 show that adaptive binarization produces a meaningful improvement in performance for our model class.

**Optimality Gap**   We solve IP (2) with a MIP solver, which finds a globally optimal solution through exhaustive search algorithms like branch-and-bound [73]. Given an instance of IP (2) and a time limit for when to stop the search, a solver returns: (i) the best feasible checklist found within the time limit – i.e., values of $\boldsymbol{\lambda}$ and $M$ that achieve the best objective value $V^{max} = l(\boldsymbol{\lambda}, M)$; and (ii) a lower bound on the objective value – i.e., $V^{min}$, the minimal objective value for any feasible solution. These quantities are used to compute the *optimality gap* $\varepsilon := 1 - \frac{V^{min}}{V^{max}}$. When the upper bound $V^{max}$ matches the lower bound $V^{min}$, the solver returns a checklist with an optimality gap of $\varepsilon = 0$ – i.e., one that is *certifiably optimal*.

In our setting, the optimality gap reflects the worst-case difference in training error between the checklist that a solver returns and the most accurate checklist that we could hope to obtain by solving IP (2). Given a solution to IP (2) with an objective value of $L$ and an optimality gap of $\varepsilon$, any feasible checklist has a training error of at least $\lceil (1 - \varepsilon)L \rceil$. When $\varepsilon$ is small, the most accurate checklist has a training error $\approx L$. Thus, if we are not satisfied with the performance of our model, we know that no checklist can perform better than $L$, and can make an informed decision to relax constraints, fit a classifier from a more complex hypothesis class, or not fit a classifier at all. When $\varepsilon$ is large, our problem may admit a checklist with training error far smaller than $L$. If we are not satisfied with the

performance of the checklist, we cannot determine whether this is because the dataset does not admit a sufficiently accurate model or because the solver has not been able to find it.

# 4 Algorithmic Improvements

In this section, we present specialized techniques to speed up the time required to find a feasible solution, and/or to produce a solution with a smaller optimality gap.

## 4.1 Submodular Heuristic

Our first technique is a submodular optimization technique that we use to generate initial feasible solutions for IP (2). We consider a knapsack cover problem. This problem can be solved to produce a checklist of $N_{\max}$ or fewer items that: (1) maximizes coverage of positive examples; (2) has a "budget" of negative examples $B$; and (3) uses at most one item from each of the feature groups $R_1, ..., R_T$ where each $R_t$ denote a set of items derived from a particular real-valued feature and $\cup_{t=1}^T R_t = [d]$.

$$\max_A \quad f_M^+(A) \tag{3a}$$
$$\text{s.t.} \quad |A| \leq N_{max} \tag{3b}$$
$$|A \cap R_t| \leq 1 \qquad t \in [T] \tag{3c}$$
$$\sum_{j \in A} c(j) \leq B \tag{3d}$$

Here, $f_M^+ := \sum_{i \in I^+} \min(\sum_{j \in A} x_{i,j}, M)$ counts the positive examples that are covered by at least $M$ items in $A$. Constraint (3b) limits the number of items in the checklist. Constraint (3c) requires the checklist to use at most one item from each feature group $R_1, \ldots, R_T$. Constraint (3d) controls the number of negative examples that are covered by at least $M$ items in $A$. This is a knapsack constraint that assigns a cost to each item as $c(j) = \sum_{i \in I^-} x_{i,j}$ and limits the overall cost of $A$ using the budget parameter $B > 0$.

The optimization problem in (3) can be solved using a submodular minimization algorithm given that $f_M^+$ is monotone submodular (see Appendix A.1 for a proof), and that constraints (3b) to (3d) are special kinds of matroid constraints. We solve (3) with a submodular minimization algorithm adapted from Badanidiyuru and Vondrák [2] (Appendix A.2). The procedure takes as input $M$, $N_{max}$ and $B$ and outputs a set of checklists that use up to $N_{max}$ items. This procedure can run within 1 second. Given a problem where we would want to train a checklist that has at most $N_{\max}$ items, we would solve this problem $|\mathcal{B}|N_{\max}$ times, varying $M \in [N_{max}]$ and $B \in \mathcal{B}$.

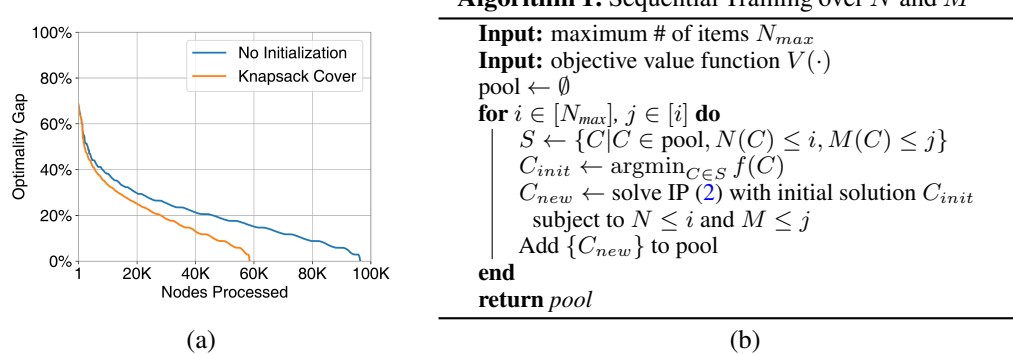

**Algorithm 1:** Sequential Training over $N$ and $M$

**Input:** maximum # of items $N_{max}$
**Input:** objective value function $V(\cdot)$
pool $\leftarrow \emptyset$
**for** $i \in [N_{max}], j \in [i]$ **do**
  $S \leftarrow \{C | C \in \text{pool}, N(C) \leq i, M(C) \leq j\}$
  $C_{init} \leftarrow \text{argmin}_{C \in S} f(C)$
  $C_{new} \leftarrow$ solve IP (2) with initial solution $C_{init}$
    subject to $N \leq i$ and $M \leq j$
  Add $\{C_{new}\}$ to pool
**end**
**return** *pool*

(a)                               (b)

**Figure 1:** (a) Performance profile when solving IP (2) with and without initialization. We show the optimality gap vs. the # of branch-and-bound nodes when solving IP (2) for the `heart` dataset where $N \leq 6$ and $M \leq 2$. (b) algorithm for sequential training over $N$ and $M$.

In Figure 1a, we show the effect of initializing the IP with checklists produced using the submodular heuristic. As shown, these solutions allow us to initialize the search with a good upper bound, allowing us to ultimately recover a certifiably optimal solution faster. Our empirical results in Section

 show that checklists from our submodular heuristic outperform checklists fit using existing baselines. This highlights the value of this approach as a heuristic to fit checklists (in that it can produce checklists that perform well), and as a technique for initialization (in that it can produce checklists that can be used to initialize IP (2) with a small upper bound).

## 4.2 Path Algorithms

Our second technique is a path algorithm for sequential training. Given an instance of the checklist IP (2), this algorithm sequentially solves smaller instances of IP (2). The algorithm exploits the fact that small instances can be solved to optimality quickly and thus "ruled out" of the feasible region of subsequent instances. The resulting approach can reduce the size of the branch-and-bound tree and improve the ability to recover a checklist with a small optimality gap. These algorithms can wrap around any procedure to solve IP (2).

In Algorithm 1, we present a path algorithm to train checklists over the full range of model sizes from 1 to $N^{\mathrm{max}}$. The algorithm recovers a solution to a problem with a hard model size limit by solving smaller instances with progressively larger limits of $N_{\mathrm{max}}$ and $M_{\mathrm{max}}$. We store the best feasible solution from each iteration in a pool, which is then used to initialize future iterations. If the best feasible solution is certifiably optimal, we add a constraint to constrict the search space for subsequent instances. This allows us to reduce the size of the branch-and-bound tree, resulting in a faster solution in each iteration.

# 5 Experimental Results

In this section, we benchmark the performance of our methods on seven clinical prediction tasks.

## 5.1 Setup

**Data.** We consider seven clinical classification tasks shown in Table 2. For each task, we create a classification dataset by using each of the following techniques to binarize features:

*Fixed*: We convert continuous and ordinal features into threshold indicator variables using the median, and convert categorical feature into an indicator for the most common category.

*Adaptive*: We convert continuous and ordinal features into 4 threshold indicator variables using quintiles as thresholds, and convert each categorical feature with a one-hot encoding.

*Optbinning*: We use the method proposed by Navas-Palencia [53] to binarize all features.

We process each dataset to oversample the minority class to equalize the number of positive and negative examples due to class imbalance.

**Methods.** We compare the performance of the following methods for creating checklists:

MIP: We fit a predictive checklist by solving an instance of IP (2) with the appropriate constraints. We solve this problem using CPLEX 12.10 [20] paired with the computational improvements in Section 4 on a 2.4 GHz CPU with 16 GB RAM for $\leq 60$ minutes.

Cover: We fit a predictive checklist using the submodular heuristic in Section 4.1. We set $N_{max} = 8$, and vary $M \in [N_{max}]$ and $B \in \{kn^+ | k \in \{\frac{1}{3}, \frac{1}{2}, 1, 2, 3\}\}$.

| Dataset | $d_T$ | $d_{pct}$ | $n$ | Prediction Task | Reference |
|---|---|---|---|---|---|
| adhd | 5 | 20 | 594 | Patient diagnosed with ADHD | [51] |
| cardio | 40 | 52 | 8,815 | Patient with cardiogenic shock died in hospital | [58] |
| kidney | 17 | 80 | 1,722 | Patient with kidney failure died after renal replacement therapy | [37] |
| mortality | 50 | 484 | 21,139 | Patient died in hospital | [37] |
| ptsd | 20 | 80 | 873 | Patient has a PCL-5 PTSD diagnosis | [51] |
| readmit | 42 | 94 | 9,766 | Patient re-admitted to hospital within 30 days | [32] |
| heart | 13 | 82 | 303 | Patient has heart disease | [24] |

**Table 2:** Clinical prediction tasks used in Section 5. Here, $n$ is the number of samples prior to oversampling, $d_T$ is the number of feature groups, and $d_{pct}$ is the number of features after binarization using the adaptive method.

| Dataset | Metric | FIXED BINARIZATION | | | | ADAPTIVE BINARIZATION | | | | LOWER BOUNDS | |
|---|---|---|---|---|---|---|---|---|---|---|---|
| | | Unit | Cover | MIP_OR | MIP | Unit | Cover | MIP_OR | MIP | LR | XGB |
| adhd $n=632$ $d_{pct}=20$ | test error (%) | 16.0 | 17.1 | 16.0 | 17.1 | 1.4 | 6.3 | 5.2 | **0.5** | 1.1 | 1.6 |
| | (min, max) | (11.9, 19.0) | (13.5, 20.6) | (11.9, 19.0) | (13.5, 20.6) | (0.8, 2.4) | (2.4, 11.7) | (4.0, 7.9) | (0.0, 0.8) | (0.8, 2.4) | (0.8, 2.4) |
| | train error (%) | 11.4 | 11.1 | 11.4 | 11.1 | 1.1 | 5.4 | 5.4 | 0.5 | **0.5** | 1.1 |
| | gap (%) | - | - | 0.0 | 0.0 | - | - | 0.0 | 0.0 | - | - |
| cardio $n=15254$ $d_{pct}=51$ | test error (%) | 23.8 | 25.3 | 29.6 | 24.1 | 23.0 | 25.3 | 29.2 | **22.6** | 21.6 | 26.3 |
| | (min, max) | (21.1, 26.5) | (21.4, 27.2) | (28.4, 31.0) | (22.5, 25.6) | (21.7, 24.9) | (23.7, 26.8) | (27.7, 30.9) | (21.5, 24.1) | (20.2, 23.3) | (25.0, 27.7) |
| | train error (%) | 25.0 | 26.2 | 29.6 | 24.1 | 23.0 | 25.5 | 29.2 | 22.5 | 20.6 | 6.4 |
| | gap (%) | - | - | 19.5 | 82.8 | - | - | 51.9 | 83.2 | - | - |
| kidney $n=1760$ $d_{pct}=80$ | test error (%) | **33.2** | 35.6 | 37.2 | 33.3 | 37.9 | 37.2 | 34.7 | 33.8 | 30.9 | 25.3 |
| | (min, max) | (33.2, 33.2) | (31.8, 38.9) | (34.7, 40.6) | (31.2, 36.1) | (36.1, 39.8) | (36.4, 38.6) | (33.2, 36.9) | (31.2, 37.5) | (28.7, 33.5) | (20.2, 28.7) |
| | train error (%) | 32.4 | 33.6 | 36.7 | 31.0 | 34.1 | 36.5 | 34.0 | 30.4 | 27.2 | 0.2 |
| | gap (%) | - | - | 45.7 | 78.9 | - | - | 43.3 | 82.4 | - | - |
| mortality $n=36684$ $d_{pct}=478$ | test error (%) | 34.3 | 38.1 | 37.2 | **29.1** | 33.8 | 36.5 | 37.8 | 29.2 | 25.0 | 28.0 |
| | (min, max) | (33.1, 36.1) | (37.4, 39.7) | (36.6, 38.4) | (27.8, 30.2) | (32.3, 35.5) | (35.5, 37.0) | (37.0, 39.4) | (29.0, 29.5) | (23.5, 26.4) | (27.2, 29.1) |
| | train error (%) | 32.9 | 36.9 | 37.4 | **29.0** | 33.4 | 36.5 | 37.8 | 29.6 | 22.6 | 4.6 |
| | gap (%) | - | - | 0.0 | 81.0 | - | - | 34.0 | 80.4 | - | - |
| ptsd $n=1106$ $d_{pct}=80$ | test error (%) | 10.9 | 10.3 | 13.8 | 11.0 | 10.2 | 12.6 | 16.2 | **8.7** | 8.0 | 4.6 |
| | (min, max) | (7.7, 14.9) | (9.1, 12.6) | (12.2, 15.8) | (6.8, 14.9) | (7.7, 12.7) | (7.7, 20.3) | (16.2, 16.2) | (5.9, 11.7) | (5.9, 10.4) | (1.8, 6.4) |
| | train error (%) | 8.3 | 9.5 | 13.4 | 8.2 | 9.2 | 10.8 | 12.5 | **5.6** | 2.8 | 0.0 |
| | gap (%) | - | - | 0.0 | 0.0 | - | - | 0.0 | 66.5 | - | - |
| readmit $n=16732$ $d_{pct}=90$ | test error (%) | 35.3 | 37.5 | 37.9 | 36.2 | 34.2 | 34.6 | 35.2 | **33.9** | 33.0 | 36.6 |
| | (min, max) | (33.8, 37.2) | (35.3, 40.0) | (36.4, 39.3) | (33.8, 37.1) | (32.6, 35.5) | (32.6, 36.2) | (34.8, 35.5) | (32.6, 35.7) | (32.2, 33.8) | (34.7, 38.1) |
| | train error (%) | 35.2 | 37.3 | 36.5 | 35.2 | 34.4 | 34.4 | 33.2 | **33.1** | 32.0 | 13.9 |
| | gap (%) | - | - | 49.3 | 81.0 | - | - | 49.9 | 73.4 | - | - |
| heart $n=330$ $d_{pct}=82$ | test error (%) | 16.3 | 18.5 | 29.1 | 28.8 | **16.1** | 23.9 | 29.1 | 16.7 | 17.6 | 17.6 |
| | (min, max) | (13.6, 24.2) | (12.1, 25.8) | (21.2, 39.4) | (28.8, 28.8) | (7.6, 31.8) | (15.2, 33.3) | (21.2, 39.4) | (13.6, 19.7) | (10.6, 27.3) | (12.1, 25.8) |
| | train error (%) | 15.2 | 16.4 | 23.6 | 13.9 | 15.2 | 18.2 | 23.6 | **13.6** | 15.2 | 9.7 |
| | gap (%) | - | - | 0.0 | 0.0 | - | - | 0.0 | 54.5 | - | - |

**Table 3:** Performance results of all methods on all datasets. For checklist models, we report the training error, test error, and optimality gap for the checklist that minimizes training error and satisfies all constraints. The intervals under test error reflect the 5-CV minimum and maximum test error. We additionally report results for LR and XGB as performance baselines that represent an informal lower bound on the error of possible checklists.

Unit: We fit a predictive checklist using unit weighting [7]. We fit $L_1$-regularized logistic regression models for $L_1$ penalties $\in [10^{-5}, 10]$. We convert each model into an array of checklists by including items with positive coefficients and the complements of items with negative coefficients. We convert each logistic regression model into multiple checklists by setting $M \in [N]$.

**Evaluation.** We use 5-fold cross validation and report the mean, minimum, and maximum test error across the five folds. We use the training set of each fold to fit a predictive checklist that contains at most $N \leq 8$ items, and that is required to select at most 1 item from a feature group. We report the training error of a final model trained on all available data.

We report results for MIP checklists where $M = 1$, which we refer to as MIP_OR since it corresponds to an OR rule. Finally, we report performance for $L_1$-regularized logistic regression (LR) and XGBoost (XGB) as baselines to evaluate performance. We train these methods using all features in an adaptive-binarized dataset to produce informal "lower bounds" on the error rate.

For each method and each dataset, we report the performance of models that achieve the lowest training error with $N \leq 8$ items. We compare the performance of *Fixed* and *Adaptive* binarization here, and report performance of *Optbinning* in Appendix D.2.

## 5.2 Results

In Table 3, we report performance for each method on each dataset when using *Fixed* binarization and *Adaptive* binarization. We report the corresponding results for *Optbinning* in Appendix D.2 due to space limitations. In Figure 2, we show the performance effects of varying $N$. Lastly, we show predictive checklists trained using our method in Figure 3 and Appendix D.3. In what follows, we discuss these results.

**On Performance** Our results in Table 3 show that our method MIP outperforms existing techniques to fit checklists (Unit) on 7/7 datasets in terms of training error, and on 5/7 datasets in terms of 5-fold CV test error. As shown in Figure 2, this improvement typically holds across all values of $N$.

In general, we find that relaxing the value of $M$ yield performance gains across all datasets (see e.g., results for ptsd where the difference in test error between MIP_OR and MIP is 7.8%), though smaller values of $M$ can produce checklists that are easier to use (see checklists in Appendix D.3).

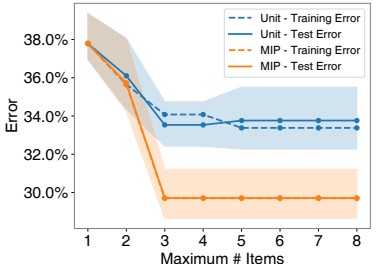

**Figure 2:** Performance of checklists with $\leq N$ items on `mortality` fit using our approach and unit weighting.

| Predict 30-day readmission if 3+ items are checked | |
|---|---|
| # admissions past year $\geq 1$ | ☐ |
| bed days $\geq 14$ | ☐ |
| length of stay $\geq 8$ days | ☐ |
| chronic or uncontrolled pain | ☐ |
| mood problems past 5 years | ☐ |
| substance abuse past month | ☐ |

**Figure 3:** Predictive checklist fit for `readmit` using MIP. The checklist has a train error of 33.1%, and a test error of 33.9%. We show sample checklists for all other datasets in Appendix D.3.

The results in Table 3 also highlight the potential of adaptive binarization to improve performance by binarizing during training time, as MIP with adaptive binarization achieves a better training error than fixed binarization for 6/7 datasets and a better test error on 5/7 datasets.

The performance loss of checklists relative to more complex model classes (LR and XGB) varies greatly between datasets. Checklists trained on `heart` and `readmit` perform nearly as well as more complex classifiers. On `mortality` and `kidney`, however, the loss may exceed 5%.

**On Generalization** Our results show that checklist performance on the training set tends to generalize well to the test set. This holds for checklists trained using all methods (i.e., Unit, Cover, MIP_OR, and MIP), which is expected given that checklists belong to a simple model class. Generalization can help us customize checklists in an informed way as we can assume that any changes to the training loss (e.g. with the addition of a constraint) lead to similar changes in the test set. In this regime, practitioners can evaluate the impact of model requirements on *predictive* performance – since they can measure differences in training error between checklists that obey different requirements, and expect these differences in training error as a reliable proxy for differences in test error.

**On the Value of an Optimality Gap** One of the benefits of our approach is that it pairs each checklist with an optimality gap, which can inform model development by identifying settings where we cannot train a checklist that is sufficiently accurate (see Section 3). On `heart`, we train a checklist with a training error of 13.6% and an optimality gap of 54.5%, which suggests that there may exist a checklist with a training error of 6.2%. This optimality gap is not available for checklists trained with a heuristic (like unit weighting or domain expertise), so we would not be able to attribute poor model performance to a difficult problem, or to the heuristic being ineffective.

**On Fairness** We analyze the fairness of MIP checklists trained on the `kidney` dataset. The task is to predict in-hospital mortality for ICU patients with kidney failure after receiving Continuous Renal Replacement Therapy (CRRT). In this setting, a false negative is often worse than a false positive, so we train checklists that minimize training FPR under a constraint to limit training FNR to 20%, using an 80%/20% train/test split.

We find that our checklist exhibit performance disparities across sex and race, which is consistent with the tendency of simpler models to exhibit performance disparities [40]. To address this issue, we train a new checklist by solving a version of IP (2) with constraints that limit disparities in training FPR and FNR over 6 intersectional groups $g \in G = \{\texttt{Male}, \texttt{Female}\} \times \{\texttt{White}, \texttt{Black}, \texttt{Other}\}$. Specifically, 30 constraints that limit the FPR disparity across groups to 15%, and 6 constraints to cap the max FNR for each intersectional group to 20%. This is unique compared to prior methods in fair classification work [76] because it enforces group fairness using exact measures[i.e., without convex relaxations 45] and over intersectional groups .

We find that we can limit performance disparities across all values of $N$ (see Figures 4 and 5, and Appendix D.4), observing that MIP with group fairness constraints has larger FPR overall but exhibits lower FPR disparities over groups (e.g., reducing the test set FPR gap between Black Females and White Males from 54.5% to 30.6%).

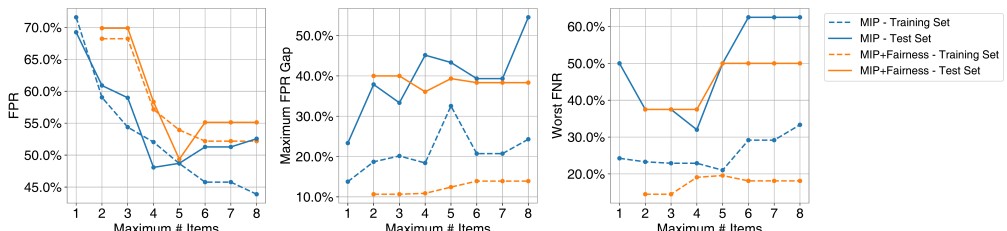

**Figure 4:** FPR (left), maximum FPR disparity (center) and worst-case FNR (right) versus number of items for checklists for mortality prediction given CRRT using `kidney`. Group fairness constraints effectively limit the FPR disparity and worst-case FNR during training, at the cost of overall performance.

| Predict Mortality Given CRRT if 3+ Items are Checked | |
|---|---|
| Age $\geq$ 66.0 years | ☐ |
| AST $\geq$ 162.6 IU/L | ☐ |
| Blood pH $\leq$ 7.29 | ☐ |
| MCV $\geq$ 99.0 fl | ☐ |
| Norepinephrine $\geq$ 0.1 mcg/kg/min | ☐ |
| Platelets $\leq$ 65.0 $\times 10^3/\mu L$ | ☐ |
| RDW $\geq$ 19.2% | ☐ |
| Time in ICU $\geq$ 14.1 hours | ☐ |

| | FNR | FPR | Worst FNR | Max FPR Gap |
|---|---|---|---|---|
| **Training** | 20.0% | 43.9% | 33.3% | 24.3% |
| **Test** | 22.2% | 52.6% | 62.5% | 54.5% |

(a) MIP

| Predict Mortality Given CRRT if 2+ Items are Checked | |
|---|---|
| ALT $\geq$ 16.0 IU/L | ☐ |
| Bicarbonate $\leq$ 17.0 mmol/L | ☐ |
| Blood pH $\leq$ 7.22 | ☐ |
| Norepinephrine $\geq$ 0.1 mcg/kg/min | ☐ |
| RDW $\geq$ 19.2% | ☐ |
| Time in ICU $\geq$ 117.3 hours | ☐ |

| | FNR | FPR | Worst FNR | Max FPR Gap |
|---|---|---|---|---|
| **Training** | 17.5% | 52.2% | 18.1% | 13.9% |
| **Test** | 19.6% | 55.1% | 50.0% | 38.3% |

(b) MIP + Fairness Constraints

**Figure 5:** Checklists trained to predict mortality given CRRT (a) without group fairness constraints and (b) with group fairness constraints. Checklist (a) has better overall FPR, but has worse FNR for the worst-case group, and exhibits disparities in FPR across groups. Checklist (b) obeys group fairness constraints that attenuate the training FPR gap to 15% and the worst-case FNR to 20%.

## 6 Learning a Short-Form Checklist for PTSD Screening

In this section, we demonstrate the practical benefits of our approach by learning a short-form checklist for Post-Traumatic Stress Disorder (PTSD) diagnosis.

**Background** The *PTSD Checklist for DSM-5* (PCL-5) is a self-report screening tool for PTSD [5, 74]. The PCL-5 consists of 20 questions that assess the severity of symptoms of PTSD DSM-5. Patients respond to each question with answers of *Not at all*, *A little bit*, *Moderately*, *Quite a bit*, or *Extremely*. Given these responses, the PCL assigns a provisional diagnosis of PTSD by counting the number of responses of *Moderately* or more frequent across four clusters of questions. Patients with a provisional diagnosis are then assessed for PTSD by a specialist. The PCL-5 can take over 10 minutes to complete, which limits its use in studies that include the PCL-5 in a battery of tests (i.e., using the provisional diagnosis to evaluate the prevalence of PTSD and its effect as a potential confounder), and has led to the development of short-form models [see e.g., 6, 43, 80].

**Problem Formulation** Our goal is to create a *short-form* version of the PCL-5 – i.e., a checklist that assigns the same provisional diagnoses as PCL-5 using only a subset of the 20 questions. We train our model using data from the AURORA study [51], which contains PCL-5 responses of U.S. patients who have visited the emergency department following a traumatic event. Here, $y_i = +1$ if the patient is assigned a provisional PTSD diagnosis. We encode the response for each question into four binary variables $x_{i,k,l} = \mathbb{1}\left[q_k \geq l\right]$ where $q_k$ is the response to each question and $l \in [4]$ is denotes its response. Our final dataset contains $d = 80$ binary variables for $n = 873$ patients, which we split into an 80% training set and 20% test set.

Since this checklist would be used to screen patients, a false negative diagnosis is less desirable than a false positive diagnosis. We train a checklist that minimizes FPR and that obeys the following operational constraints: (1) limit FNR to 5%; (2) pick one threshold for each PCL-5 question; (3) use at most 8 questions. We make use of the same methods and techniques as in Section 5.2.

**Results**   Our results show that short-form checklists fit using our method outperform other checklists (see Figure 6). In Figure 7, we display a sample $N = 8, M = 4$ checklist trained with our method which reproduces the PCL-5 diagnosis with high accuracy, offering an alternative to the PCL-5 with simpler binary items and less than half the number of questions.

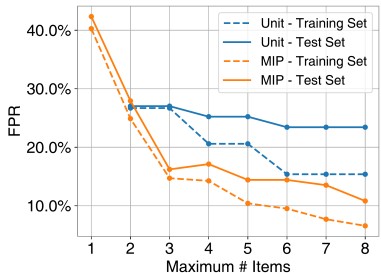

| Predict PTSD if 4+ Items are Checked | |
| --- | --- |
| Avoiding thinking about experience $\geq$ moderately | ☐ |
| Avoiding activities or situations $\geq$ moderately | ☐ |
| Blame for stressful experience $\geq$ a little bit | ☐ |
| Feeling distant or cut off $\geq$ quite a bit | ☐ |
| Irritable or angry outbursts $\geq$ moderately | ☐ |
| Loss of interest in activities $\geq$ moderately | ☐ |
| Repeated disturbing dreams $\geq$ moderately | ☐ |
| Trouble experiencing positive feelings $\geq$ moderately | ☐ |

**Figure 6:** FPR versus number of items for short-forms of the PCL-5. MIP outperforms Unit across all values of $N$.

**Figure 7:** Short-form of the PCL-5 trained with MIP. The checklist has a train/test FPR of 6.6%/10.8%, and a train/test FNR of 4.7%/6.3%. It reproduces the PCL-5 diagnosis on the training/test sets with 94.1/90.9% accuracy, and has a 70.8% optimality gap. We show condensed PCL-5 questions due to space constraint.

## 7   Concluding Remarks

In this work, we presented a machine learning method to learn predictive checklists from data by solving an integer program. Our method illustrates a promising approach to build models that obey constraints on qualities like safety [1] and fairness [3, 14]. Using our approach, practitioners can potentially co-design checklists alongside clinicians – by encoding their exact requirements into the optimization problem and evaluating their effects on predictive performance [17, 36].

We caution against the use of items in a checklist as causal factors of outcomes (i.e., questions in the short-form PCL are not "drivers" of PTSD), and against the use of our method solely due to its interpretability. Recent work has argued against the use of models simply because they are "interpretable" [25] and highlights the importance of validating claims on interpretability through user studies [42, 59].

We emphasize that effective deployment of the checklist models we create does not end at adoption. The proper integration of our models in a clinical setting will require careful negotiation at the personal and institutional level, and should be considered in the context of care delivery [28, 62, 63] and clinical workflows [26, 55].

## Acknowledgments and Disclosure of Funding

This material is based upon work supported by the National Science Foundation under Grant No. 2040880. We acknowledge a Wellcome Trust Data for Science and Health (222914/Z/21/Z) and NSERC (PDF-516984) grant. Resources used in preparing this research were provided, in part, by the Province of Ontario, the Government of Canada through CIFAR, and companies sponsoring the Vector Institute. Dr. Marzyeh Ghassemi is funded in part by Microsoft Research, and a Canadian CIFAR AI Chair held at the Vector Institute. We would like to thank Taylor Killian, Bret Nestor, Aparna Balagopalan, and four anonymous reviewers for their valuable feedback.

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
