# A  Submodular Optimization Algorithms

## A.1  Proof of Monotonicity and Submodularity

In Equation (3a), we stated the objective of the knapsack cover to be

$$f_M^+(A) = \sum_{i \in I^+} \min\left(\sum_{j \in A} x_{i,j}, M\right)$$

defined over a ground set $\Omega$. Here, we prove that it is monotone submodular.

**Remark 1.** $f_M^+$ *is monotonically increasing.*

**Proof.**  Let $e \in \Omega$ and $A' := A \cup \{e\}$. Since $x_{i,j} \in \{0,1\}$, $\sum_{j \in A'} x_{i,j} \geq \sum_{j \in A} x_{i,j} \ \forall \ i \in I^+$, and thus $f_M^+(A') \geq f_M^+(A)$. $\qquad\square$

**Remark 2.** $f_M^+$ *is submodular. i.e.*

$$\forall A \subseteq B \subseteq \Omega, \ \forall e \in \Omega, \ f_M^+(B \cup \{e\}) - f_M^+(B) \leq f_M^+(A \cup \{e\}) - f_M^+(A)$$

**Proof.**

$$
\begin{aligned}
&f_M^+(B \cup \{e\}) - f_M^+(B) \\
&= \sum_{i \in I^+} \min\left(\sum_{j \in B \cup \{e\}} x_{i,j}, M\right) - \min\left(\sum_{j \in B} x_{i,j}, M\right) \\
&= \sum_{i \in I^+} x_{i,e} \mathbb{I}\left(\sum_{j \in B} x_{i,j} < M\right) && \text{Since } x_{i,e}, x_{i,j} \in \{0,1\} \text{ and } M \in \mathbb{N} \\
&\leq \sum_{i \in I^+} x_{i,e} \mathbb{I}\left(\sum_{j \in A} x_{i,j} < M\right) && \text{Since } A \subseteq B \text{ and } x_{i,j} \in \{0,1\} \\
&= f_M^+(A \cup \{e\}) - f_M^+(A) && \square
\end{aligned}
$$

## A.2  Knapsack Cover

To find a solution to problem 3, we use the greedy algorithm proposed by Badanidiyuru and Vondrák [2], which deals with submodular maximization subject to a system of $l$ knapsack constraints and with $p$ matroid constraints. We present an adapted version of the algorithm in Algorithm 2 where $l = 1$. Here, $p = 2$ if both the maximum item constraint (i.e. cardinality matroid) and the one item per feature group constraint (i.e. partition matroid) are enforced. The $\epsilon$ parameter allows us to

trade-off solution time and solution quality. In this work, we set $\epsilon = 0.2$. This algorithm yields a $1/(p + 3 + \epsilon)$ approximation ratio [2].

---

**Algorithm 2:** Greedy Knapsack Cover (one knapsack constraint and $p$ matroid constraints).

**Input:** feature set $A$, submodular function $f : 2^A \rightarrow \mathbb{R}_+$, oracle for $p$-system $\mathcal{I}$, budget $B \in \mathbb{R}_+$, cost function $c : A \rightarrow [0, B]$, step size $\epsilon > 0$

**Output:** A set of possible solutions $\mathcal{S}$, with each solution $S \in \mathcal{S}$, $S \subseteq A$, satisfying $S \in \mathcal{I}$ and $\sum_{j \in S} c(j) \leq B$

1: $n \leftarrow |A|$
2: $m \leftarrow \max_{j \in A} f(j)$
3: $\mathcal{S} \leftarrow \emptyset$
4: **for** $\rho \leftarrow \frac{m}{p+1}, (1+\epsilon)\frac{m}{p+1}, ..., \frac{2nm}{p+1}$ **do**
5:     $\tau \leftarrow \max\{f(j) : \frac{f(j)}{c(j)/B} \geq \rho\}$
6:     $S \leftarrow \emptyset$
7:     **while** $\tau \geq \frac{\epsilon m}{n}$ *and* $\sum_{j \in S} c(j) \leq B$ **do**
8:        **for** $j \in A$ **do**
9:           $\delta \leftarrow f(S \cup \{j\}) - f(S)$
10:           **if** $S \cup \{j\} \in \mathcal{I}$ *and* $\delta \geq \tau$ *and* $\frac{\delta}{\sum_{j \in S} c_j/B} \geq \rho$ **then**
11:             $S \leftarrow S \cup \{j\}$
12:             **if** $\sum_{j \in S} c(j) > B$ **then**
13:                $\mathcal{S} \leftarrow \mathcal{S} \cup (S \setminus \{j\})$
14:                $\mathcal{S} \leftarrow \mathcal{S} \cup \{j\}$
15:                **goto** 4
16:             **end**
17:           **end**
18:        **end**
19:        $\tau \leftarrow \frac{1}{1+\epsilon}\tau$
20:     **end**
21: **end**
22: **return** $\mathcal{S}$

---

# B Sequential Training Algorithms

**Error Path** To learn optimal checklists for a problem subject to $FNR \leq FNR_{max}$ while minimizing FPR, we can sequentially train checklists with an increasingly larger $FNR$ constraint, using all previously trained checklists as initial solutions. This allows us to obtain an array of checklists across the ROC curve. We present this algorithm in Algorithm 3.

---

**Algorithm 3:** Sequential Training with $FNR$ constraint

**Input:** $FNR_{max}$, grid width $\epsilon$, loss function $f$
pool $\leftarrow \emptyset$
**for** $i \leftarrow \epsilon, 2\epsilon, ..., FNR_{max}$ **do**
    $S \leftarrow \{C | C \in \text{pool}, FNR(C) \leq i\}$
    $C_{init} \leftarrow \text{argmin}_{C \in S} f(C)$
    $C_{new} \leftarrow$ solve Formulation (2) with initial solution $C_{init}$ subject to $N \leq i$, potentially with Algorithm 1
    pool $\leftarrow$ pool $\cup \{C_{new}\}$
**end**
**return** *pool*

---

## C  Hyperparameter Grid for Baselines

Here, we describe the hyperparameter grids for the lower bound baselines shown in Table 3. For LR, we use L1 regularized logistic regression from the Scikit-Learn library [56], using the liblinear optimizer and varying $C \in \{10^{-4}, 10^{-3.5}, ..., 10^1\}$. For XGB, we use the XGBoost library [15], varying the maximum depth $\in \{1, 2, ..., 6\}$ and setting all other hyperparameters at default values.

## D  Supporting Material for Experimental Results

### D.1  Datasets

All datasets used in this paper (i.e. in Table 2) are publicly available, with the exception of `readmit`. Datasets based on MIMIC-III [37] (`kidney`, `mortality`) and eICU [58] (`cardio`) are hosted on PhysioNet under the PhysioNet Credentialed Health Data License[1]. The ADHD and PTSD datasets are from the AURORA study [51], which is hosted on the National Institute of Mental Health (NIMH) data archive, subject to the NIMH Data Use Agreement[2]. The `heart` dataset is hosted on the UCI Machine Learning Repository under an Open Data license. In cases where data access requires consent or approval from the data holders, we have followed the proper procedure to obtain such consent. All datasets used in this study have been deidentified and contain no offensive content. We briefly describe each dataset and preprocessing steps taken below.

**adhd**  We use data from the attention deficit hyperactivity disorder (ADHD) questionnaire contained within the AURORA study [51], which consists of of U.S. patients who have visited the emergency department (ED) following a traumatic event. It consists of five questions selected from the Adult ADHD Self-Report Scale (ASRS-V1.1) Symptom Checklist[3] (specifically, questions 1, 9, 12, 14, and 16), answered on a 0-4 ordinal scale (i.e. 0 = *never*, 1 = *rarely*, 2 = *sometimes*, 3 = *often*, 4 = *very often*). The target is the patient's clinical ADHD status. This results in a dataset containing 594 patients with a prevalence of 46.8%.

**cardio**  Cardiogenic shock is a serious acute condition where the heart cannot provide sufficient blood to the vital organs. Using the eICU Collaborative Research Database V2.0 [58], we create a cohort of patients who have cardiogenic shock during the course of their intensive care unit (ICU) stay using an exhaustive set of clinical criteria based on the patient's labs and vitals (i.e. presence of hypotension and organ hypoperfusion). The goal is to predict whether a patient with cardiogenic shock will die in hospital. As features, we summarize (minimums and maximums) relevant labs and vitals (e.g. systolic BP, heart rate, hemoglobin count) of each patient from the period of time prior to the onset of cardiogenic shock up to 24 hours. This results in a dataset containing 8,815 patients, 13.5% of whom die in hospital.

**kidney**  Using MIMIC-III and MIMIC-IV [37], we create a cohort of patients who were given Continuous Renal Replacement Therapy (CRRT) at any point during their ICU stay. For patients with multiple ICU stays, we select their first one. We define the target as whether the patient dies during the course of their selected hospital admission. As features, we select the most recent instances of relevant lab measurements (e.g. sodium, potassium, creatinine) prior to the CRRT start time, along with the patient's age, the number of hours they have been in ICU when CRRT was administered, and their Sequential Organ Failure Assessment (SOFA) score at admission. We treat all variables as continuous with the exception of the SOFA score, which we treat as ordinal. This results in a dataset of 1,722 CRRT patients, 51.1% of which die in-hospital. We define protected groups based on the patient's sex and self-reported race and ethnicity.

**mortality**  We follow the cohort creation steps outlined by Harutyunyan et al. [34] for their in-hospital mortality prediction task. We select the first ICU stay longer than 48 hours of patients in MIMIC-III [37], and aim to predict whether they will die in-hospital during their corresponding hospital admission. As features, we bin the time-series lab and vital measurements provided by

---

[1]https://physionet.org/content/mimiciii/view-license/1.4/
[2]https://nda.nih.gov/ndapublicweb/Documents/NDA+Data+Access+Request+DUC+FINAL.pdf
[3]https://add.org/wp-content/uploads/2015/03/adhd-questionnaire-ASRS111.pdf

Harutyunyan et al. [34] into four 12-hour time-bins, and compute the mean in each time-bin. We additionally include the patient's age and sex as features. This results in a cohort of 21,139 patients, 13.2% of whom die in hospital.

**ptsd**  We use data from the PTSD questionnaire contained within the AURORA study [51], which consists of U.S. patients who have visited the emergency department following a traumatic event. It consists of responses to all items on the PTSD Checklist for DSM-5 (PCL-5), which are answered on a 0-4 ordinal scale (i.e. 0 = *not at all*, 1 = *a little bit*, 2 = *moderately*, 3 = *quite a bit*, 4 = *extremely*). To obtain the PCL-5 diagnosis, we use the DSM-5 diagnostic rule [5], which assigns a positive diagnosis to those with a *moderately* or higher on at least: 1 Criterion B item (questions 1-5), 1 Criterion C item (questions 6-7), 2 Criterion D items (questions 8-14), 2 Criterion E items (questions 15-20). This results in a dataset containing 873 patients with a prevalence of 36.7%.

**heart**  We use the Heart dataset from the UCI Machine Learning Repository, where the goal is to predict the presence of heart disease from clinical features. It consists of 303 patients, 54.5% of which have heart disease. We use all available features, treating *cp*, *thal*, *ca*, *slope* and *restecg* as categorical, and all remaining features as continuous.

**readmit**  The readmit dataset involves predicting 30-day hospital readmission using features derived from natural language processing on clinical records at the Massachusetts General Hospital in Boston, Massachusetts. Further details of the dataset can be found in Greenwald et al. [32], and we have obtained permission to use this dataset from the authors. Note that we only use data from Massachusetts General Hospital, which consists of 9,766 samples with a 14.3% prevalence. We treat the bed days, the number of prior admissions, and the length-of-stay as continuous, and all other variables as categorical.

## D.2  Additional Experimental Results

We compare the performance of adaptive binarization versus Optbinning in Table 4 on a subset of the datasets. We find that neither procedure consistently outperforms the other.

| | Training Error | | Test Error | | Optimality Gap | |
|---|---|---|---|---|---|---|
| | **Adaptive** | **Optbinning** | **Adaptive** | **Optbinning** | **Adaptive** | **Optbinning** |
| kidney | 30.4% | **29.0%** | 33.5% (31.3%, 36.4%) | **33.1%** (31.5%, 36.4%) | 82.4% | 79.3% |
| mortality | **29.7%** | 34.5% | **29.7%** (28.7%, 31.2%) | 36.0% (34.1%, 38.2%) | 75.1% | 100.0% |
| readmit | 33.1% | **33.0%** | 34.3% (33.3%, 35.2%) | **33.8%** (33.2%, 34.5%) | 73.4% | 82.4% |
| heart | 13.6% | **11.5%** | 18.2% (16.7%, 19.7%) | **15.2%** (15.2%, 15.2%) | 54.5% | 45.4% |

**Table 4:** Error rates and optimality gaps of checklists trained using MIP for a variety of checklists binarized using adaptive and optbinning. Confidence bounds for the test error correspond to minimum and maximum test errors from 5-fold CV.

## D.3 Sample Checklists

For each dataset, we show sample checklists created from adaptive-binarized data using MIP_OR and MIP. For each method and dataset, we show the checklist with the lowest training error in Figure 8.

| Predict ADHD if 1+ Items are Checked | |
| --- | --- |
| trouble wrapping up final details $\geq 4$ | ☐ |
| difficulty concentrating $\geq 2$ | ☐ |
| leave seat in meetings $\geq 2$ | ☐ |
| difficulty unwinding and relaxing $\geq 2$ | ☐ |

**(a)** Checklist fit for `adhd` using MIP_OR, with train error = 5.4%, test error = 5.2% (4.0%, 7.9%), optimality gap = 0.0%.

| Predict Mortality if 1+ Items are Checked | |
| --- | --- |
| MET | ☐ |
| Min heart rate $\geq 100$ | ☐ |
| Min respiratory rate $\geq 25$ | ☐ |
| Min SpO2 $\leq 88$ | ☐ |

**(c)** Checklist fit for `cardio` using MIP_OR, with train error = 29.2%, test error = 29.2% (27.7%, 30.9%), optimality gap = 51.9%.

| Predict Mortality Given CRRT if 1+ Items are Checked | |
| --- | --- |
| Bicarbonate $\leq 14.0$ | ☐ |
| Platelets $\leq 65.0$ | ☐ |
| Norepinephrine $\geq 0.1003$ | ☐ |

**(e)** Checklist fit for `kidney` using MIP_OR, with train error = 34.0%, test error = 34.7% (33.2%, 36.9%), optimality gap = 43.3%.

| Predict In-Hospital Mortality if 1+ Items are Checked | |
| --- | --- |
| 36h-48h: Glascow coma scale total mean $\leq 14.17$ | ☐ |

**(g)** Checklist fit for `mortality` using MIP_OR, with train error = 37.8%, test error = 37.8% (37.0%, 39.4%), optimality gap = 34.0%.

| Predict PTSD if 1+ Items are Checked | |
| --- | --- |
| avoiding thinking about experience $\geq 2$ | ☐ |
| trouble remembering stressful experience $\geq 4$ | ☐ |
| loss of interest in activities $\geq 4$ | ☐ |
| irritable or angry outbursts $\geq 4$ | ☐ |
| blame for stressful experience $\geq 4$ | ☐ |

**(i)** Checklist fit for `ptsd` using MIP_OR, with train error = 12.5%, test error = 16.2% (16.2%, 16.2%), optimality gap = 0.0%.

| Predict 30-Day Readmission if 1+ Items are Checked | |
| --- | --- |
| Jail past 5 years | ☐ |
| Maximum care past year $= 1$ | ☐ |
| Poor competency past 5 years $= 1$ | ☐ |
| State care past 5 years $= 1$ | ☐ |
| Bed days $\geq 3.0$ | ☐ |
| Length of stay $\geq 8.0$ | ☐ |

**(k)** Checklist fit for `readmit` using MIP_OR, with train error = 33.2%, test error = 35.2% (34.8%, 35.5%), optimality gap = 49.9%.

| Predict ADHD if 2+ Items are Checked | |
| --- | --- |
| trouble wrapping up final details $\geq 2$ | ☐ |
| difficulty concentrating $\geq 2$ | ☐ |
| leave seat in meetings $\geq 2$ | ☐ |
| difficulty unwinding and relaxing $\geq 2$ | ☐ |
| finishing sentences of other people $\geq 2$ | ☐ |

**(b)** Checklist fit for `adhd` using MIP, with train error = 0.5%, test error = 0.5% (0.0%, 0.8%), optimality gap = 0.0%.

| Predict Mortality if 3+ Items are Checked | |
| --- | --- |
| Mechanical Ventilation | ☐ |
| Min heart rate $\geq 100$ | ☐ |
| Min systolic BP $\leq 80$ | ☐ |
| Max respiratory rate $\leq 12$ | ☐ |
| Min respiratory rate $\geq 20$ | ☐ |
| Min SpO2 $\leq 88$ | ☐ |
| Max anion gap $\geq 14$ | ☐ |
| Max BUN $\geq 25$ | ☐ |

**(d)** Checklist fit for `cardio` using MIP, with train error = 22.5%, test error = 22.6% (21.5%, 24.1%), optimality gap = 83.2%.

| Predict Mortality Given CRRT if 3+ Items are Checked | |
| --- | --- |
| Bicarbonate $\leq 17.0$ | ☐ |
| AST $\geq 174.0$ | ☐ |
| RDW $\geq 19.2$ | ☐ |
| Norepinephrine $\geq 0.300$ | ☐ |
| Time in ICU $\geq 29.32$ | ☐ |
| Age $\geq 66.0$ | ☐ |
| MCV $\geq 99.0$ | ☐ |

**(f)** Checklist fit for `kidney` using MIP, with train error = 30.4%, test error = 33.8% (31.2%, 37.5%), optimality gap = 82.4%.

| Predict In-Hospital Mortality if 2+ Items are Checked | |
| --- | --- |
| 36h-48h: Fraction inspired oxygen mean measured | ☐ |
| 12h-24h: Glascow coma scale total mean measured | ☐ |
| 36h-48h: Glascow coma scale total mean $\leq 14.17$ | ☐ |
| 36h-48h: Mean blood pressure mean not measured | ☐ |

**(h)** Checklist fit for `mortality` using MIP, with train error = 29.6%, test error = 29.2% (29.0%, 29.5%), optimality gap = 80.4%.

| Predict PTSD if 4+ Items are Checked | |
| --- | --- |
| repeated disturbing dreams $\geq 2$ | ☐ |
| avoiding thinking about experience $\geq 2$ | ☐ |
| avoiding activities or situations $\geq 2$ | ☐ |
| trouble remembering stressful experience $\geq 2$ | ☐ |
| loss of interest in activities $\geq 1$ | ☐ |
| feeling distant or cut off $\geq 2$ | ☐ |
| irritable or angry outbursts $\geq 3$ | ☐ |
| blame for stressful experience $\geq 2$ | ☐ |

**(j)** Checklist fit for `ptsd` using MIP, with train error = 5.6%, test error = 8.7% (5.9%, 11.7%), optimality gap = 66.5%.

| Predict 30-Day Readmission if 3+ Items are Checked | |
| --- | --- |
| Mood problems past 5 years $= 1$ | ☐ |
| Substance abuse past month $= 0$ | ☐ |
| Bed days $\geq 14.0$ | ☐ |
| # admissions past year $\geq 1.0$ | ☐ |
| Length of stay $\geq 8.0$ | ☐ |

**(l)** Checklist fit for `readmit` using MIP, with train error = 33.1%, test error = 33.9% (32.6%, 35.7%), optimality gap = 73.4%.

```
┌─────────────────────────────────────────────────┐
│ Predict Heart Disease if 1+ Items are Checked    │
│ cp ! = 0                                       □  │
└─────────────────────────────────────────────────┘
```

**(m)** Checklist fit for `heart` using MIP_OR, with train error = 23.6%, test error = 29.1% (21.2%, 39.4%), optimality gap = 0.0%.

```
┌─────────────────────────────────────────────────┐
│ Predict Heart Disease if 4+ Items are Checked    │
├─────────────────────────────────────────────────┤
│ sex = 0                                        □  │
│ cp ! = 0                                       □  │
│ chol ≤ 255.4                                   □  │
│ oldpeak ≤ 1.92                                 □  │
│ slope ! = 1                                    □  │
│ ca = 0                                         □  │
│ thal = 2                                       □  │
└─────────────────────────────────────────────────┘
```

**(n)** Checklist fit for `heart` using MIP, with train error = 13.6%, test error = 16.7% (13.6%, 19.7%), optimality gap = 54.5%.

**Figure 8:** Checklists with the lowest training error created from adaptive-binarized data using MIP_OR and MIP on each dataset. We show condensed items for `adhd` and `ptsd` due to space limitations.

### D.4 Additional Fairness Results

In Table 5, we show the training and test FNR and FPR for each subgroup corresponding to the intersection of race and sex on `kidney` for the checklists shown in Figure 5.

| Protected Group | Method | Train FNR | Train FPR | Test FNR | Test FPR |
|---|---|---|---|---|---|
| **White M** | MIP | 18.3% | 50.2% | 21.3% | 45.5% |
| | MIP + Fairness | 17.9% | 54.5% | 22.7% | 52.7% |
| **White F** | MIP | 18.7% | 35.2% | 16.2% | 46.0% |
| | MIP + Fairness | 18.0% | 50.3% | 16.2% | 52.0% |
| **Black M** | MIP | 33.3% | 42.9% | 62.5% | 70.0% |
| | MIP + Fairness | 12.5% | 42.9% | 37.5% | 70.0% |
| **Black F** | MIP | 15.2% | 59.5% | 25.0% | 100.0% |
| | MIP + Fairness | 12.1% | 56.8% | 50.0% | 83.3% |
| **Other M** | MIP | 21.0% | 38.4% | 16.7% | 50.0% |
| | MIP + Fairness | 18.1% | 50.5% | 8.3% | 45.0% |
| **Other F** | MIP | 23.2% | 46.3% | 28.0% | 73.3% |
| | MIP + Fairness | 17.9% | 55.6% | 16.0% | 66.7% |

**Table 5:** FNR and FPR for the intersections of race and sex on the `kidney` dataset, for our MIP method with and without group fairness constraints.