# OpenReview forum: "Learning Optimal Predictive Checklists"
_NeurIPS.cc/2021/Conference — NeurIPS 2021 Poster_

### Official Review · Reviewer_rvfW · 2021-07-13

**Rating:** 9
**Confidence:** 4

**Summary:**

This paper proposed a formulation and learning algorithms for learning diagnostic checkslists.  A checklist is a set of binary items where the prediction is positive if the number of checked items exceeds a threshold.

The paper proposes two techniques to solve the resultant integer program optimization problem.  The first is a submodular heuristic
The second is a sequential algorithm that reduces the size of the branch and bound tree by eliminating checklists that would be subsets of checklists that do not perform well.

The submission includes evaluation on multiple datasets and under different binarization schemes. For one dataset, there is an evaluation of group-wise bias in the predictions based on the learned lists. There is also a more detailed case study for PTSD that demonstrates how to translate a realistic goal into the setup needed for the learning problem.

**Ethical Concerns:**

No ethical concerns

**Limitations And Societal Impact:**

The authors address the social impact of their work very well, throughout the paper.  The motivation in the introduction makes it clear.  The authors evaluate for bias in one dataset and show how their framework allows for a context-specific mitigation to reduce the most important gaps in performance. The conclusion includes specific cautions for the interpretation and application of their work both with respect to fairness and its role as an interpretable model.

**Main Review:**

# Originality
This work provides a formulation for a task that has not been adopted in machine learning, but is highly relevant to healthcare.

# Quality
This work is thorough and well described in the submission. The experiments support the technique's performance boradly and suggest where future improvements may lie.  The case study on PTSD works very well to illustrate how to take a variety of real-world goals and formulate a checklist search problem.



# Clarity
This paper is overall understandable.  There are minor clarity issues that make some sections require multiple attempts or later context to understand, but they do not interfere with understanding the main conclusions or methods.

The abstract doesn't stand well on its own.  After reading the paper it reads like a good summary, but before reading, too many parts read as undefined terms.

Section 4.2 is a little hard to parse. Parts of it read as if it's the introduction to a forthcoming more detailed description of the algorithm, but there is no longer description section.

The fairness evaluation is less strong than the rest of the paper and could be improved some. It's not clear how "a false negative is worse than a false positive" (line249) supports minimizing FPR and holding FNR as a constraint and evaluating the FPR gap. This may be a clarity issue, so please explain that connection more clearly, otherwise, your setup seems to suggest that the FNR should be monitored more closely. If false negatives are worse than false positives, it would seem to follow that a gap in the FNR would also be worse- or does setting it as a constraint also make the FNR equal across groups?



# Significance
This work represents a highly interpretable model that presents the predictive task in a similar form to what clinicians already use.  Checklists are widely adopted and trusted in medicine for many reasons and learning the from data presents an opportunity to improve real practice. The performance as is with the current adaptive binarization is comparable to more complex learning algorithms.


# Minor Issues


- line 22, is it that there are no standardized *automated* approaches or truly no standards in how checklists are developed? expert panels often have strict protocols that they use that do serve as a type of of standardization, though it's less strict than computerized standardization
- line 47 uses PCL without it being defined yet.
- line 71: "reduce computation" here is not clear, a more specific phrase would help. Because that part is talking about interpretable models, computation could refer to either at the time of learning or prediction.
- Table 1 is difficult to read small
- section heading 4.2, should that be plural or singular?
- line 171: "The algorithms exploits" -> "The algorithm exploits"
- line 188: "... by using one.." to "...by using each..." as  it reads, I was looking for how you chose which one to use until I saw the results
- the table 3 caption should state that the "lower bound" columns are a baseline and the way they're referred to in the table should match the text (line 212)
- on line 225 it says the checklists performs similarly to more complex classifiers, this could use a citation,
- line 279, define ED

**Time Spent Reviewing:**

3

---

> ### Author Response · Authors · 2021-08-10
> **Response to Reviewer rvfW**
>
> Thank you for the detailed review and constructive suggestions!
>
> ### Fairness Constraints in Section 5.2
>
>
> Thank you for bringing this to our attention! This was actually a real-world application where the constraints and preferences were set forth by clinical collaborators. To be sure, the language in Section 5.2 does reflect what they asked and what we did – but we now see that language needs to be revised to explain this in greater detail.
>
> In short, we were still working under the assumption that a false negative is worse than a false positive. That being said, the key requirement was to enforce a "per-group FNR'' limit as a “worst-case” fairness guarantee (e.g., a kind of Rawlsian fairness guarantee between intersectional groups). In this case, our collaborators were clear that their goal was not to equalize FNR across groups - since this might have led to unacceptable levels of FNR for all groups.
>
>
> ### Automated Approaches for Checklist Deployment
>
> There are indeed no standard automated approaches for checklist creation, but there are processes and guidelines for checklist development by expert panels. In our literature review, however, we found that the development of checklists could vary substantially across applications, even in the same domain [e.g. 1-3].
>
> ### Abstract
>
> We will clarify some of the terms in the abstract.
>
> ### Path Algorithm
>
> We will add a more detailed description of the path algorithm in Section 4.2.
>
> ### Minor Issues
>
> Thank you for pointing these out. We will fix them in our revised version.
>
> ### References
> [1]  Stufflebeam, Daniel L. "Guidelines for developing evaluation checklists: the checklists development checklist (CDC)." Kalamazoo, MI: The Evaluation Center Retrieved on January 16 (2000): 2008.
>
> [2] Hales, Brigette, et al. "Development of medical checklists for improved quality of patient care." International Journal for Quality in Health Care 20.1 (2008): 22-30.
>
> [3] Ogden, Shanna R., et al. "Developing a checklist: consensus via a modified Delphi technique." Journal of cardiothoracic and vascular anesthesia 30.4 (2016): 855-858.

---

> > ### Comment · Reviewer_rvfW · 2021-08-24
> > **thanks for clarifications**
> >
> > those answer my questions and minor revisions should help clarify.

---

### Official Review · Reviewer_rAcQ · 2021-07-16

**Rating:** 6
**Confidence:** 4

**Summary:**

Method to train classifiers based on integer programing optimization of binarized variables.
Evaluate and multiple public healthcare related datasets, comparing accuracy vs. other baseline methods.
Offers some optimality gap guarantees of models chosen within time constraints (vs. exhaustive search).


**Ethical Concerns:**

No major concerns, as largely just describing a risk score learning method.
Offers a somewhat perfunctory "fairness" sub-study on example model performance for different racial subgroups.
An interesting demonstration of the trade off between more "fair" performance characteristics across racial subgroups, but resulting with worse performance on average for all. This is a very brief aside study on one example dataset however, so not enough to really explore the issue in a meaningful way within the scope of this paper.


**Limitations And Societal Impact:**

As above, no major concerns beyond existing tools and technology, but also not clear a substantial difference/impact beyond existing ones.


**Main Review:**

Seems to be confusing the term "checklist" here. What is described are more analogous to clinical risk scores (see MDCalc.com for implemented example).
This is quite different than the checklists "widely used to promote safety and reliability" eluded to in the Intro such as surgical pre-op checklists. Most of those are about consistently executing a process and have nothing to do with prediction, risk assessment, diagnosis, or calculation.

The construction and evaluation of the test models developed overall seem fair, but as eluded to in the Intro, these seem to function very similarly to many existing methods for sparse linear classifiers with small integer coefficients. Neither based on performance or interpretability did there seem to be a dramatic advantage to the proposed approach.
The "certifiably optimal result" was an interesting contribution to define an optimality gap between the current best performing model vs. best conceivably possible given the available data variables.

Not clear there's much value in reporting train set performance throughout manuscript. Is mostly distracting and even confusing as I first thought the XGBoost models were far outperforming the other models, but that's just because they were overfit to the training data. Potential generalizable accuracy on the test sets is much more relevant.


**Time Spent Reviewing:**

2

---

> ### Author Response · Authors · 2021-08-10
> **Response to Reviewer rAcQ**
>
> Thank you for the detailed review and constructive suggestions!
>
> ### Predictive vs. Procedural Checklists
>
> We agree that we are not trying to learn procedural checklists (such as surgical checklists) from data, but rather we are trying to learn predictive models from data that have a similar format as surgical checklists. Admittedly, this point has become somewhat less clear than we had intended as a result of several revisions, and we plan to make it clear in the manuscript.
>
> Specifically, our view is that many predictive models can promote beneficial outcomes in clinical settings, but ultimately fail to do so due to disuse (that itself stems from, e.g., distrust or high deployment costs). One strategy to overcome these issues is to try and build a checklist first. Checklists will not perform well for all classification tasks, but it's worth trying because they sometimes can (as we show in this work). In applications where checklists do perform well, we have good reason to believe that the models will overcome the hurdles that affect adoption. This is motivated by the widespread adoption of checklists in clinical settings. You are right in that checklists that are adopted in clinical settings lead to different kinds of benefits and succeed due to different mechanisms. That being said, they are ultimately adopted because their format makes them easy to use and easy to understand.
>
> ### Goals of the Fairness Demo
>
> We think that we may have failed to articulate the goals of this section. Our goal in including this application in the paper was to show that our approach can handle fairness constraints in a way that is seamless and flexible, rather than to provide an in-depth analysis of fairness for the particular problem. We think that this is valuable since fairness is an important consideration for clinical prediction models [1], and because our approach can handle fairness constraints in a way that is considerably more powerful than state-of-the-art techniques (i.e., we can enforce them between intersectional subgroups, and without regularization). We will add some guiding text to make this clearer in the contributions and in the section header itself.
>
>
> ### Reporting Out-of-Sample Performance
>
> We agree that test set performance is more important than training set performance. We report training errors throughout the paper since it reflects the objective function of the MIP, and allows us to check if checklists are, in fact, generalizing (see also a comment by Reviewer hq2w). We will make this clearer in the revision.
>
> ### Comparison with Existing Methods
>
> We agree that there are other methods to train sparse linear classifiers with small integer coefficients, and we do cite them in our paper. Our goal in this work is to adapt this family of approaches to design a specialized technique to train the best possible checklists. In addition to the presence of the optimality gap, one of the benefits of this was that we could easily design specialized techniques that would not apply to the broad family of models with integer coefficients (e.g. the submodular heuristic). Our paper also proposes some routines that could be applied to more general methods (e.g., the path algorithms). Lastly, it highlights valuable functionality that are missing in previous methods (e.g., the ability to address fairness constraints as discussed above).
>
>
> We hope that our response resolves any misunderstandings! Thank you again for your time, and we look forward to resolving any remaining concerns that you may have.
>
> ### References
>
> [1] Vyas, D. A., Eisenstein, L. G., & Jones, D. S. (2020). Hidden in plain sight—reconsidering the use of race correction in clinical algorithms.

---

> > ### Comment · Reviewer_rAcQ · 2021-08-31
> > **Emphasize distinctions from other sparse learners**
> >
> > My assessments are largely the same, but I'll move to marginal acceptance.
> > I'm not sure this line of work functionally offers much more than existing sparse learners, but increased emphasis on the distinguishing elements like estimating optimality gaps may offer useful content for the community.
> > I would avoid referencing surgical pre-op checklists and similar in the Intro, as those serve a completely different type of (process) function than the score / risk calculation the authors are proposing here.

---

> > > ### Author Response · Authors · 2021-08-31
> > > **Thank you**
> > >
> > > Thank you for your valuable feedback, and for re-evaluating the score. We will emphasize and clarify the use of the optimality gap (in addition to our response to Reviewer hq2w), as well as remove mentions of procedural checklists from the introduction.

---

### Official Review · Reviewer_hq2w · 2021-07-16

**Rating:** 7
**Confidence:** 3

**Summary:**

This work presents an extensive evaluation of checklist models (linear models with binary features and coefficients) for clinical predictions.  Such models are quite transparent relative to more complex families, and can be potentially be extremely simple to implement in clinic.  The learning problem is framed as mixed integer programming, which has other virtues - solvers can provide bounds on how far our current solution is to the optimal solution wrt training set performance, and the inclusion of desirable properties as constraints.  This work also provides an evaluation of binarization methods for continuous variables (recall that features must be binary), and proposes a pair of methods for improving performance of the learned models.  Extensive evaluations are performed on 8 datasets; the checklist models very often achieve quite good performance on par with more complex model families.

**Limitations And Societal Impact:**

The authors are commendably straight forward about some of the technical limitations of their work but it could also be pointed out that the path to clinical utility is a bit more complicated than printing out a checklist and posting it on exam room walls - effective adoption does not end at deployment.




**Main Review:**

This work presents a study of a very restricted class of linear models - binary features and coefficients - for clinical prediction tasks.  Such models are basically checklists - you evaluate some predicates and add up the true values and see if the sum >= some threshold.  Such models have many virtues - the chief ones being that they are potentially very simple to deploy - you can print them out, and clinicians can run them with minimal arithmetic (though evaluating the predicates can still be non-trivial). They are also transparent in the sense of how they arrive at scores (it should be noted that this is not the only important sense of transparent - it does not, as the authors point out, imply causal mechanisms - but lack of this sort of transparency can sometimes be a barrier to adoption).  The formulation finding optimal checklists as a mixed integer program admits the use of off the shelf solvers, and the encoding of some desirable properties (e.g., keeping the number of predicates small, bounding the difference in performance between subgroups small) as constraints on the optimization.  It can also provide bounds on the gap between the current solution and the best possible solution with respect to training set error.  The authors also propose two methods that are claimed to enable better solutions (with respect to time required and quality).  Evaluations are performed on 8 prediction tasks, each using a different dataset, and the checklist models are for the most part on par with some more complex model classes suggesting plenty of room for trading off marginal gains in performance vs ease of use and deployment, etc.

Overall, I liked this work a lot and _really_ appreciated the clarity of the writing.  I am not aware of previous uses of MIP for finding optimal checklist models for clinical prediction, though of course there is plenty of prior work (NB - this work was cited appropriately by the current work) on linear models with integer coefficients, etc.  Although this work does not present much very new in the way of core methods - it does provide a useful evaluation of a far simpler model class, and the benefits admitted by framing it as an MIP; I think it would be useful for the NeuRIPS community, particularly the subset focused on clinical prediction tasks, to see this work.  There are, however, some ways in which the work could be improved.

First, I strongly recommend doing an ablation of the sub-modularity heuristic and the path algorithm presented in sections 4.1 and 4.2.  The authors demonstrate the benefit of the adaptive binarization scheme (this scheme is not novel but does seem to have a quite substantial effect on bringing the performance of the checklist models on par with the other baselines!) but not these two components of their method.

Second, in order for the lasso and xgb baselines to be trusted as reasonable lower bounds on error, it is important to provide some detail on hyperparameter tuning.  It doesn't have to be a lot, and can reasonably be relegated to supplementary materials as long as readers are directed there in the main manuscript.

Third, I recommend being a bit more circumspect in the language describing the utility of the optimality gaps.  These are bounds regarding the training set behavior, not generalization error.  The authors argue that the extremely simple model class prevents large divergences between training and generalization performance most of the time, and this is borne out for the most part in the evaluations.  Perhaps the authors could emphasize how precisely the gap would be used in practice (i.e., as they implied, to help guide the developer in, say, relaxing constraints on number of predicates, etc, to achieve better performance)?

In summary, I think this is a very clearly written, well organized paper presenting a useful perspective on how to build clinically useful diagnostic or risk prediction models. There are some gaps in the evaluation I would like to see filled.




**Time Spent Reviewing:**

4

---

> ### Author Response · Authors · 2021-08-10
> **Response to Reviewer hq2w**
>
> Thank you for the detailed review and constructive suggestions!
>
>
> ### Ablation Study
>
> Thanks for this suggestion! We agree that the ablation study will help pinpoint the performance gains associated with our methodological improvements. We will include this experiment in our revision. In general, we would expect to see larger optimality gaps in the ablations similar to Figure 1a, especially for larger datasets.
>
>
> ### Details on Hyperparameter Grid
>
> Thank you for pointing this out. We ran our experiments using the following hyperparameter grid for these methods:
> - L1-regularized LR: regularization strength $\in$ { $10^{-4}, 10^{-3.5}, …, 10^1$ }
> - XGB: max depth $\in$ { $1, 2, …, 6$ }
>
>
> We will include a description of this in the text – either in the experimental section or the supplementary material based on space constraints. We are happy to expand the hyperparameter grid if need be.
>
>
> ### Language on Optimality Gaps:
>
> We agree with your takeaway and will revise our language to make it clear that the optimality gap only provides bounds on the training error, and does not imply that the model is optimal with respect to test error. In practice, we expect that performance on the training set should carry over into "good" performance on the test set (which could theoretically be quantified using generalization bounds). In practice, we think that practitioners should always verify that the checklists do in fact generalize (e.g., by comparing training error to an estimate of out-of-sample error via a hold-out set or via K-fold CV).
>
> As for how the optimality gap would be used in practice, we offer the following scenarios:
> - If a model with constraints does not perform well and has a large optimality gap, increasing the training time can only improve training accuracy.
> - If, at any point, a model offers a lower bound on the training error (computed using the optimality gap) that is not satisfactory, then training should be stopped and constraints should be relaxed (e.g. increasing $N_{max}$).
> - If a model without constraints does not perform well, then this indicates that no good checklist exists, and a different model class should be used.
>
> Please let us know if our explanation does not adequately address the concern, and we will gladly follow up. We plan to include a discussion about this in Section 3 (where the formally introduce the optimality gap) as well as in Section 6 (where we work with constraints).
>
>
>
> ### On Effective Adoption
>
> You are quite right that effective adoption does not end at deployment! This comment led to a long and interesting discussion on our end, as well as the discovery of work in this area that highlights the importance of clinical integration [see e.g., 1, 2, 3]. We now plan to include a short discussion about this in Section 7. We appreciate any other pointers you may have.
>
> ### References
> [1] Elish, Madeleine Clare. "The stakes of uncertainty: developing and integrating machine learning in clinical care." Ethnographic Praxis in Industry Conference Proceedings. Vol. 2018. No. 1. 2018.
>
> [2] Shah, Pratik, et al. "Artificial intelligence and machine learning in clinical development: a translational perspective." NPJ digital medicine 2.1 (2019): 1-5.
>
> [3] Sendak, Mark P., et al. "A path for translation of machine learning products into healthcare delivery." EMJ Innov 10 (2020): 19-172.

---

> > ### Comment · Reviewer_hq2w · 2021-08-24
> > **Response(response)**
> >
> > Dear Authors,
> >
> > Thank you very much for your response, and apologies for the tardiness of my response.  In brief, I am happy with your responses.  As for Effective Adoption, if you are not already aware of the line of work about Kaiser Permanente's early warning system for in-patient deterioration, it may be worth tracing - very little was published about the model itself; a lot was published about the roll out of the workflow, design of the workflow, etc...  Some of it is very specific to roll out to a rather large healthcare system, much of the rest may still be interesting to you. A good starting point is Paulson 2020, What Do We Do After the Pilot Is Done? Implementation of a Hospital Early Warning System at Scale.
> >
> > Best regards,
> >
> > Reviewer hq2w

---

> > > ### Author Response · Authors · 2021-08-25
> > > **Thank you for your response**
> > >
> > > Thank you for pointing us to this line of work! It is definitely relevant to our discussion of clinical adoption, and we will reference [e.g. 1-3] as an example of the careful design and development of clinical workflow and governance structure required to successfully integrate ML models into the clinical setting.
> > >
> > > [1] Dummett, B. Alex, et al. "Incorporating an early detection system into routine clinical practice in two community hospitals." Journal of hospital medicine 11 (2016): S25-S31.
> > >
> > > [2] Escobar, Gabriel J., et al. "Automated identification of adults at risk for in-hospital clinical deterioration." New England Journal of Medicine 383.20 (2020): 1951-1960.
> > >
> > > [3] Paulson, Shirley S., et al. "What do we do after the pilot is done? Implementation of a hospital early warning system at scale." The Joint Commission Journal on Quality and Patient Safety 46.4 (2020): 207-216.

---

### Official Review · Reviewer_Xd4U · 2021-07-17

**Rating:** 6
**Confidence:** 3

**Summary:**

The paper proposed a new method to learn predictive clinical checklists for classification tasks by solving an integer programming problem.The method was validated and evaluated with 8 benchmark clinical datasets with consideration of ML fairness.

**Ethics Review Area:**

["I don’t know"]

**Limitations And Societal Impact:**

Yes.

**Main Review:**

The paper proposed an interesting method to learn a short-form checklists from clinical datasets with the potential application of clinical diagnosis, treatment decision making, etc. The paper showed clear problem constructions, originality of the work with comparison to related literatures and decent considerations on model fairness as well as social impact.
Technical analysis might require more experiments to improve the soundness and confirm the utilities of the proposed method. Detailed questions/suggestions are as follows:
1. As indicated in Authors' checklist, no confidence intervals are reported for experiment results. Considering the oversampling on minority class, boostrapping on training data might be necessary to obtain confidence intervals to evaluate performance robustly. Cross validation could also be applied to test generalization ability of different models.
2. The paper chose accuracy as the main performance comparison metric across different datasets, which could be optimized considering the real-world scenarios. However, depending on the nature of the task, the cost of false positive (FP) and the cost of false negative (FN) prediction might vary a lot. e.g. For PTSD diagnosis, missing a single positive case might lead to severe outcome in delayed treatment, FNR at a fixed TPR might be a better evaluation metric. But for the mortality task, TPR given a fixed FPR is a more important metric since we need to be very cautious in decision-making on withdrawal of life supporting treatment when a ML classifier makes prediction of "positive" in mortality.
3. A fundamental question: For each of the classification task, feature selection and feature importance could be easily obtained, which could be used as a potential checklist as well. What will be the advantages of the proposed method with comparison to get checklists from feature selection?

**Time Spent Reviewing:**

1

---

> ### Author Response · Authors · 2021-08-10
> **Response to Reviewer Xd4U**
>
> Thank you for the detailed review and constructive suggestions!
>
> ### Reporting Confidence Intervals
>
> We agree that reporting confidence intervals for test errors would be more informative. We have started running experiments to produce these estimates, and plan to include them in an updated version of Table 3 for the revision. Based on the experiments that we have completed, we expect that our main takeaways in Section 5 will remain the same. We expect to see variations in test performance across all methods – with higher variance on the datasets with smaller sample sizes. We also expect to observe higher variance for the heuristics/baseline methods as they may fail for some folds of the data and not others.
>
> ### Training Checklists that Strike a Balance with FPR/FNR
>
>
> We agree that it is important to train checklists that are responsive to different costs on false positives and false negatives in practice.
>
> We only reported accuracy as the main performance metric for the experiments in Section 5.1, since that section was meant to benchmark our methods against the baselines. In this case, we wanted to report accuracy on a "rebalanced dataset" since it would let us make meaningful comparisons of methods across datasets (i.e., comparisons of model performance for a specific combination of FN/FP costs, and that are unlikely to produce a "trivial classifier" that predicts the  majority class).
>
> We evaluate methods in terms of class-based accuracy metrics (e.g., TPR/FPR) in the demonstrations in Sections 5.2 and 6 for the reasons you brought up. Our intention was that these sections would demonstrate how the method would be used in real-world applications and so it was important to train checklists that were responsive to differential costs for FPs and FNs. Looking back, we think that this could have been stated more clearly - and so we will consider bundling Sections 5.2 and 6 together as "Demonstrations."
>
> One note: our method actually provides two different ways to handle classification problems with different costs on FP/FN:
> - We can have users specify misclassification costs for FPs/FNs in the objective function (in this case, it has the benefit that costs represent actual costs of FP/FN since we use the zero-one loss).
>
> - We can have users specify limits on FPR and FPR – e.g., return the checklist that "maximizes training TPR subject to training FPR $\leq$ 20%."
>
> In the demonstrations in Section 5.2 and 6, we currently use the second approach since it showcases a less common approach to handling these settings (i.e., the ability for users to specify hard constraints on FPR/FNR). We are happy to also showcase the alternative approach if you think it would be useful. In either case, we plan to include a short discussion about this point in the text.
>
> ### Building Checklists using Important Features
>
> Yes! There is a large literature in statistics and the social sciences on techniques to create simple checklist-like classifiers by adding together "important" binary features (see e.g. [1-3]).
>
> This family of techniques is typically called "Unit Weighting" or the "Burgess Method." Broadly speaking, most techniques could be used to produce models with the same form as our checklists, and there is considerable variation between techniques in the way that they binarize and identify "important" features.
>
> The "UnitWeighting" method in Sections 5 and 6 is meant to demonstrate some of the characteristics of this family of methods. In this case, the "important" features are those chosen by L1 regularized logistic regression. We are happy to include other variations that may appear interesting.
>
> Overall, our finding is that "unit weighting" methods can work, but that their performance is not reliable. When the methods fail to produce a checklist that performs well (as they sometimes do), we cannot tell if it is because the prediction problem is hard or because the technique has somehow failed. This shortcoming is especially problematic in settings where checklists must adhere to constraints – as most methods need to be changed in ad hoc ways to handle these requirements.
>
> ### References
>
> [1] Dawes, Robyn M. "The robust beauty of improper linear models in decision making." American psychologist 34.7 (1979): 571.
>
> [2] Einhorn, Hillel J., and Robin M. Hogarth. "Unit weighting schemes for decision making." Organizational behavior and human performance 13.2 (1975): 171-192.
>
> [3] Bobko, Philip, Philip L. Roth, and Maury A. Buster. "The usefulness of unit weights in creating composite scores: A literature review, application to content validity, and meta-analysis." Organizational Research Methods 10.4 (2007): 689-709.

---

### Author Response · Authors · 2021-08-10
**Thank you for your constructive comments!**

We thank all reviewers for their time and feedback! We were glad to see that our work was positively received, and that reviewers described it as “highly relevant to healthcare” (Reviewer rvfW), “very clearly written” (Reviewer hq2w), with “clear problem constructions” (Reviewer Xd4U). We also appreciate their questions, comments, and suggestions. We provide answers to the main points raised by each reviewer below, and outline changes that we plan to address in a potential revision. Please feel free to follow up with us! We very much welcome any feedback that can further strengthen the paper.

---

### Public Comment · ~Alexander_Berman2 · 2023-08-21
**Unclear origin of dataset**

When trying to understand the checklist for the "heart" dataset (figure 8n), I noticed that the value for at least one of the features (thal=2) is outside the set of values in the official dataset (3, 6 and 7). Also, in the supplementary material the authors write that the disease prevalence in the dataset is 54.5%, while in the official dataset the prevalance is 46%. These observations suggest that the authors have used another version of the dataset than what is stated in the paper.

---

### Decision · Program_Chairs · 2021-09-27

**Decision:**

Accept (Poster)

**Comment:**

This work proposes an IP-based solution to learning predictive clinical checklists. Overall, the paper was well received, but reviewers make a number of recommendations to help improve the paper that should be incorporated prior to presentation:
-including error bars/confidence intervals
-including an ablation of the sub-modularity heuristic and the path algorithm
-compare to the strongest baselines possible (thorough baseline hyperparameter turning)
-framing of fairness results
I encourage authors to carefully consider reviewer feedback when working on their revision.